# The Effect of Emotional Labor on the Physical and Mental Health of Health Professionals: Emotional Exhaustion Has a Mediating Effect

**DOI:** 10.3390/healthcare11010104

**Published:** 2022-12-29

**Authors:** Chien-Chih Chen, Yu-Li Lan, Shau-Lun Chiou, Yi-Ching Lin

**Affiliations:** 1Department of Future Studies and LOHAS Industry, Fo Guang University, Yilan 262307, Taiwan; 2Department of Health Administration, Tzu Chi University of Science and Technology, Hualien 970302, Taiwan; 3Department of Rehabilitation, Taiwan Adventist Hospital, Taipei 10556, Taiwan

**Keywords:** emotional labor, emotional exhaustion, physical and mental health, medical professional technicians

## Abstract

(1) Background: Workers who perform emotional labor for an extended period are prone to emotional exhaustion; in particular, when the work exceeds the range of one’s emotional resources, it will produce job burnout. This study investigated the effects of emotional labor and emotional exhaustion on the physical and mental health of health professionals. (2) Methods: This study was cross-sectional and the sampling criteria were health professionals from August 2020 to July 2021, including rehabilitators, nutritionists, clinical psychologists, radiologists, respiratory therapists, pharmacists, medical examiners and audiologists. A questionnaire was used to collect data on participants’ emotional labor, emotional exhaustion, physical health and mental health. A total of 120 valid questionnaires were obtained. (3) Results: Significant positive correlations were found between emotional labor and emotional exhaustion, physical and mental health and anxiety. A hierarchical regression analysis found that the effect of emotional labor on physical and mental health increased the predictive power to 59.7% through emotional exhaustion, and emotional exhaustion had a mediating effect on the relationship between emotional labor and physical and mental health. (4) Conclusions: This study provides a reference for managers of medical institutions to care for employees’ work stress and physical and mental health, which will help institutions build a friendly and healthy workplace.

## 1. Introduction

As the population ages rapidly, the need for chronic disease care and the severity of disease increases. In addition to having a wealth of knowledge and experience in managing ever-changing medical technologies, medical staff also need to deal with hospital evaluations and maintain a good doctor–patient relationship. Their work performance behaviors can have a massive impact on patient outcomes and safety [1].

Healthcare work involves elevated levels of physical and psychosocial stress, resulting in low employee job satisfaction, burnout, turnover intentions and poor health [2,3,4,5]. In particular, during the COVID-19 pandemic, fear rose in medical staff and caused more health problems to surface, which not only had a huge impact on their psychological well-being, but also tested their resilience and ability to cope with stress. Therefore, the psychological and physical effects of COVID-19 on medical personnel have been the focus of attention [6].

The object of medical care services is the patient. During the extensive medical service process, it is necessary for health professionals to suppress their true feelings, so they are often required to maintain a good attitude and manage their emotions to provide quality healthcare services. Emotional labor is an individual’s commitment to the management of emotions to create appropriate facial expressions and body movements in front of the public [7]. Emotional labor is divided into two levels of expression: surface and deep. Surface acting involves masking actual emotions such as using a fake smile to hide one’s true feelings, whereas deep acting involves trying to feel and express desired emotions such as modifying one’s feelings to suit the situation [7,8].

Empirical studies on emotional labor have found that the frequency of deep emotional performance of nursing staff correlates with higher emotional labor and a less ideal state of mental health [9]. Workers who perform emotional labor for an extended period are prone to emotional exhaustion [10,11]; in particular, when the work exceeds the range of one’s emotional resources, it will produce job burnout [12].

If employees have negative emotions or cannot control their emotions rationally, and support and methods to eliminate emotional problems are not available, employees may experience job burnout and reduced job performance, which can result in emotional exhaustion [13]. When emotional exhaustion occurs, employees will feel that they are emotionally disconnected; their physical and emotional energy are exhausted, and they have negative emotions such as anxiety, tension, depression and irritability. Moreover, they will display discontentment toward work and lose their jobs. Lack of interest and enthusiasm [14,15], and then symptoms such as physical problems, lack of energy, deteriorating health and exhaustion follow [16]. Therefore, emotional exhaustion is one of the important indicators of physical and mental health [15].

Previous studies have demonstrated that emotional labor directly affects organizational outcomes and employee well-being [17,18,19,20,21,22]. However, few studies have investigated the relationship between emotional labor and physical and mental health. As for medical personnel, past research has focused on the emotional labor of nursing staff, and research on the effects of emotional labor and emotional exhaustion on physical and mental health of health professionals does not exist. Therefore, this study is expected to provide a reference for managers of medical institutions to care for employees’ work stress and physical and mental health, which will help institutions build a friendly and healthy workplace.

## 2. Materials and Methods

### 2.1. Study Design and Participants

This study was cross-sectional and used purposive sampling. From August 2020 to July 2021, questionnaires were distributed to health professionals, including rehabilitators, nutritionists, clinical psychologists, radiologists, respiratory therapists, pharmacists, medical examiners and audiologists. The eligible sampling conditions included health professionals working in the hospital who were willing to participate in the study as indicated by their signed consent. The exclusion criteria included those employees who had submitted their resignation. A total of 120 valid questionnaires were obtained. In terms of gender, women and men accounted for 80% (96) and 20% (24), respectively. Ages 30–40 accounted for 41.7% (50), followed by 40–50 at 27.5% (33). More than half (56.7%, *n* = 68) of the participants were married, and 41.7% (50) were unmarried. Those without children were the majority, accounting for 54.2% (65), followed by those with two children, accounting for 26.7% (32). Most participants had a university degree, accounting for 67.5% (81), or had completed graduate school or more, accounting for 20.0% (24). In terms of occupation, rehabilitator accounted for 24.2% (29), followed by medical examiners, accounting for 20% (24) (Table 1).

### 2.2. Measures

A structured questionnaire was used as the research tool. Five experts and scholars were invited to evaluate the content validity of the questionnaire. They rated each item on the questionnaire with regard to “importance,” “text clarity,” and “appropriateness.” The reliability was analyzed using Cronbach’s alpha. The questionnaire included the following measures. Emotional labor items were rated using a five-point Likert scale [23], which included 7 items on surface acting and 4 items on deep acting. After factor analysis, two eigenvalues greater than 1 were extracted, the eigenvalues of a single factor were 3.42 and 4.12, respectively, and their sum explained 71.37% of the variance. The factor loading of each item is greater than 0.6, indicating that the item validity is good (Table 2). The reliability of deep effect and surface effect were 0.904 and 0.893, respectively, the Cronbach’s alpha of emotional labor was 0.885, and the content validity was 0.881.

Emotional exhaustion was assessed using the Maslach Burnout Inventory (MBI-GS) [24] which contains five items that are responded to using a five-point Likert scale, with a good-enough reliability of 0.88. In this study, Cronbach’s alpha was 0.919 and content validity was 0.933. In terms of physical and mental health, the China Health Questionnaire (CHQ-12) revised by Williams [25] in 1986 was used. On the basis of the 30 questions in the General Health Questionnaire (GHQ) [26], various questions about Chinese culture were added. The Cronbach coefficient of the scale was 0.83–0.92. The measure included 4 questions about physical health, 4 questions about anxiety, and 4 questions about depression and poor family relationships. The Cronbach’s alpha of this study was 0.839 and content validity was 0.890.

### 2.3. Data Collection

After the study was approved by the Taiwan Adventist Hospital Institutional Review Board (108-E-21), the researcher explained the purpose of the study and relevant information to eligible participants. After the participant signed the written consent, they put their completed questionnaire and consent form into a secured box that was allocated for questionnaires.

### 2.4. Differences in Physical and Mental Health according to Demographic Characteristics

Differences in physical and mental health due to gender (t = 0.358, *p* > 0.05) and marital status (t = −0.298, *p* > 0.05) were evaluated using a t-test, and there were no significant differences. One-way analysis of variance was used to explore the differences in physical and mental health according to age (F = 0.291, *p* > 0.05), education (F = 0.691, *p* > 0.05), and profession (F = 1.658, *p* > 0.05). There were no statistically significant differences found among the groups.

## 3. Result

### 3.1. Descriptive Analysis for Each Scale

Table 3 shows the results for the average scores on the measures. The average score for emotional labor was 3.70 (1–5 points). The average score for surface acting was 3.42, and the average score for deep acting was 4.11. The average score for emotional exhaustion was 2.71 (1–5 points). Cutoff scores for the above were as follows: low group < 1.35, middle group 1.35–3.65, high group > 3.65. The average score for physical and mental health was 2.01 (1–4 points). Among them, the average scores were 1.72 for physical condition, 1.87 for anxiety, and 2.38 for depression and poor family relations. The average score for poor sleep was 2.13. Cutoff scores for the above were as follows: low group < 1.08, middle group 1.08–2.92, high group > 2.92.

Therefore, the health professional’s surface acting is moderate, deep acting is high, emotional exhaustion is moderate, poor physical condition is moderate, anxiety is moderate, depression and poor family relationships are moderate, and poor sleep is moderate.

### 3.2. Differences in Emotional Labor, Emotional Exhaustion, and Physical and Mental Health among Health Professionals

Kruskal–Wallis analysis was used to explore the differences in emotional labor (including surface acting and deep acting), emotional exhaustion, and physical and mental health in the different professions. The results show that surface acting (*p* < 0.05) and emotional exhaustion (*p* < 0.05) were significantly different among the health professionals categories (F = 3.491, *p* < 0.05).

The surface acting of the examiner and audiologist is the highest, and the surface acting of the clinical psychologist is the lowest. Emotional exhaustion was highest for examiner and pharmacist and lowest for radiologist and nutritionist.

### 3.3. The Relationship between Emotional Labor, Emotional Exhaustion, and Physical and Mental Health

Pearson correlation coefficients were used to analyze the relationships between the variables. Significant positive correlations were found between emotional labor and emotional exhaustion (r = 0.336, *p* < 0.001) and physical and mental health (r = 0.184, *p* < 0.05). The more surface acting (r = 0.918, *p* < 0.001) and deep acting (r = 0.659, *p* < 0.001), the more emotional labor. Emotional labor was also significantly positively correlated with anxiety (r = 0.225, *p* < 0.05), indicating that the greater the degree of anxiety, the more emotional labor. Emotional labor, physical condition, and depression and poor family relations did not reach a statistically significant difference, indicating that physical condition as well as depression and poor family relations have no correlation with emotional labor. Emotional exhaustion was significantly positively correlated with physical and mental health (r = 0.597, *p* < 0.001) and surface acting (r = 0.404, *p* < 0.001). This shows that the worse the physical and mental health and the more surface acting, the greater the emotional exhaustion. Deep acting and emotional exhaustion did not reach a statistically significant difference, indicating that there is no correlation between them. Emotional exhaustion was significantly positive with physical condition (r = 0.482, *p* < 0.001), anxiety (r = 0.596, *p* < 0.001), and depression and poor family relations (r = 0.294, *p* < 0.001). This shows that the worse the physical condition, the more anxiety and the more depression and poor family relations, the greater the degree of emotional exhaustion. Physical and mental health, surface acting and deep acting did not reach a statistically significant difference, indicating that there is no correlation between them. Physical and mental health and physical condition (r = 0.857, *p* < 0.001), anxiety (r = 0.878, *p* < 0.001), and depression and poor family relations (r = 0.600, *p* < 0.001) had a significantly positive correlation. This means that the worse the physical condition, the more anxiety, and the more depression and poor family relations, the worse the physical and mental health (Table 4).

### 3.4. The Mediating Effect of Emotional Exhaustion on the Relationship between Emotional Labor and Physical and Mental Health

A hierarchical regression analysis was used to explore emotional exhaustion as a mediator in the relation between emotional labor and physical and mental health. The results show that Model 1 (F = 4.138, *p* < 0.05), Model 2 (F = 14.996, *p* < 0.001) and Model 3 (F = 32.447, *p* < 0.001) were all statistically significant. The *R*^2^ of emotional labor on physical and mental health was 18.4% (β = 0.184, *p* < 0.05), and the *R*^2^ of emotional exhaustion on physical and mental health was 33.6% (β = 0.336, *p* < 0.001). The effect of emotional labor on physical and mental health increased the predictive power to 59.7% through emotional exhaustion, and emotional exhaustion had a mediating effect on the relationship between emotional labor and physical and mental health (β = 0.603, *p* < 0.001) (Table 5).

## 4. Discussion

This study found that the higher the emotional labor of health professionals, the greater the emotional exhaustion. Similar to the discussion by scholars such as Hochschild [7], when employees are required to regulate their emotions at work to match the changes in the external environment, the appropriate external emotional performance may be inconsistent with the employee’s true inner feelings; the external and internal emotions are alienated. This alienation cannot be maintained for a long time, which leads to stress overload and emotional exhaustion. Therefore, workers with high emotional labor are more likely to have emotional disorders and emotional exhaustion. Given the correlation between emotional labor and emotional exhaustion [27], the higher the emotional labor undertaken by an individual, the more there will be an increase in the frequency of emotional exhaustion [28].

The results of this study show that surface acting was positively correlated with emotional exhaustion, while deep acting had no significant relation to emotional exhaustion. The plausible reason is that surface acting only modifies the external emotional expression to achieve the purpose of emotional camouflage, which would make it easy to create conflict between inner and external emotions, resulting in emotional imbalance. Therefore, the higher the level of surface acting, the heavier the emotional labor required, which is akin to the view put forward by Grandey [29,30]. When an individual’s inner emotions are repressed and desired emotions are expressed, the surface effects exhibited can deplete personal energy and affect employee well-being [30,31,32]. For example, nurses are more likely to experience emotional exhaustion when they engage in superficial performances [33]. Surface behavior is positively correlated with emotional exhaustion [31,34]. Conversely, deep performances tend to resolve the initial emotional dissonance, resulting in the same internal feelings and external performance [30,31]. Employees will experience positive emotional experiences to express positive emotions, and these experiences may provide relief of fatigue [35]. A prior study found that deep acting leads to better mood, better job performance and higher job satisfaction [29].

Long-term and constant emotional labor may damage an individual’s physical and mental health [36,37]. This study found that the emotional labor of health professionals was significantly and positively correlated with anxiety and physical and mental health. Previous studies also found that surface acting emotional labor was positively associated with depression [38,39]. A study on nursing staff found that emotional labor can explain 21% of the variance in mental health status, showing that the greater the emotional labor of nursing staff, the less ideal was their mental health [9]. Additionally, a recent study found that episodic emotional labor was a strong predictor of depressive symptoms in nursing home health care workers two years later [40].

Past studies have discovered that in the process of performing medical care, nurses must suppress their emotions to provide professional services. Such emotional services can lead to work stress, deterioration of physical and mental health and emotional exhaustion [41], and directly affect the quality of patient care and the occurrence of medical negligence, which can even prompt them to leave the workplace [42]. Emotional labor can predict employee job satisfaction and emotional exhaustion. Nursing staff who use superficially disguised emotional labor are more likely to experience emotional exhaustion and have lower job satisfaction because they need to hide their true emotions and disguise unfelt emotions, increasing the degree of emotional dysregulation [43].

Moreover, the present study observed that surface acting emotional labor and emotional exhaustion were positively correlated, and the latter had a mediating effect on the relation between emotional labor and physical and mental health. Similar to previous studies, nurses are more likely to experience emotional exhaustion when they use surface acting emotional labor. As a way of regulating emotions, performances can help employees regain emotional resources and reduce emotional exhaustion [44]. In addition, Rogers et al. [38] found positive correlations between the surface acting emotional labor of doctors and work-related burnout and depression. There was a negative correlation between deep emotional labor and burnout, and work-related burnout mediated the relationship between surface emotional labor and depression.

The above literature suggests that the higher the level of emotional labor, the more serious the degree of emotional exhaustion, and the less ideal one’s physical and mental health. Conversely, it also indicates the lower the emotional labor, the lower the degree of emotional exhaustion and the better one’s physical and mental health. Therefore, it is inferred that there will be significant differences between emotional labor and emotional exhaustion and physical and mental health of different degrees.

This study adopted a cross-sectional data collection method to explore the emotional labor, emotional exhaustion, and physical and mental health of health professionals. It is suggested that future research conduct longitudinal studies to examine causal relationships and effects over time. In addition, the study included health professionals at one particular hospital, and there may be different results due to differences in the background variables of the research participants.

## 5. Conclusions

This study confirms that the emotional labor and emotional exhaustion of health professionals are quite serious. Emotional labor was significantly positively correlated with emotional exhaustion, physical and mental health, and anxiety. Emotional exhaustion was significantly and positively correlated with physical and mental health, physical condition, anxiety, and depression and poor family relationships. The predictive power of emotional labor on physical and mental health was improved through emotional exhaustion; emotional exhaustion had a mediating effect in the relation between emotional labor and physical and mental health.

We hope to inspire others to study how emotion management can help the health and well-being of healthcare professionals, thereby improving the quality of care healthcare professionals provide to patients. Therefore, developing emotional management skills necessary for health professionals to work effectively is essential to improve the quality of patient care and treatment outcomes, and to ensure that patient care is not compromised by the health professional’s own emotional and health conditions. There is indeed a great need for channels to relieve and de-escalate their emotional labor to prevent emotional exhaustion from occurring. Related studies have found that social support through colleagues, supervisors and organizations in the workplace may reduce the negative effect of emotional labor [45,46]. Therefore, the implications of this study suggest that in addition to improving the working environment (including providing social support) to reduce the emotional labor of employees’ surface acting behavior, hospital managers can provide supportive psychological counseling to reduce employees’ emotional labor and emotional exhaustion. This in turn will improve the physical and mental health and well-being of employees.

## Figures and Tables

**Table 1 healthcare-11-00104-t001:** Descriptive statistics of participants (*n =* 120).

Variable	Item	*n*	(%)	Variable	Item	*n*	(%)
Gender	Male	24	20	Number of children	None	65	54.2
	Female	96	80		One	19	15.8
Age	20–30	15	12.5		Two	32	26.7
	31–40	50	41.7		Three or more	4	3.3
	41–50	33	27.5	Professional category	Rehabilitator	29	24.2
	51–60	17	14.2		Nutritionist	5	4.2
	61–70	5	4.2		Clinical psychologist	8	6.7
Education	College	15	12.5		Radiologist	20	16.7
	University	81	67.5		Respiratory therapist	8	6.7
	Master’s degree	24	20		Pharmacist	23	19.2
Marital status	Unmarried	50	41.7		Medical examiner	24	20
	Married	68	56.7		Audiologist	3	2.5
	Divorce	1	0.8				
	Widowed	1	0.8				

**Table 2 healthcare-11-00104-t002:** Factor analysis results of emotional labor (*n* = 120).

Emotional Labor	Varimax Factor One Factor Loadings	Varimax Factor Two Factor Loadings
**Surface acting**		
I will feign appropriate emotion when serving patients.	0.800	
I will pretend to be in a good mood when serving patients.	0.854	
I treat patients like a play.	0.852	
When serving patients, I will feign job-appropriate emotions.	0.879	
In order to express the emotions required by the job, I will wear a mask to hide what I really feel inside.	0.842	
There is a gap between the emotions I express to the patient and what I feel inside.	0.621	
I would feign appropriate emotions to treat patients.	0.836	
**Deep acting**		
I try to feel the emotions that need to be expressed when serving patients.		0.809
I try to empathize with the emotions that have to be expressed in serving patients.		0.880
I try to feel the emotions that need to be expressed to the patient.		0.898
I try my best to serve patients with empathy.		0.843
Percentage of variation explained (%)	49.87	21.50
KMO (Kaister–Meyer–Olkin)		0.880
Bartlett’s test of sphericity		χ^2^ = 920.927
		*p* < 0.001

**Table 3 healthcare-11-00104-t003:** Descriptive analysis for each scale (*n* = 120).

Measurement Constructs/Items	Number of Items	Mean	Item Score Range
Emotional labor	11	3.70	1–5
Surface acting	7	3.42	1–5
Deep acting	4	4.11	1–5
Emotional exhaustion	5	2.71	1–5
Physical and mental health	12	2.01	1–4
Physical condition	4	1.72	1–4
Anxiety	3	1.87	1–4
Depression and poor family relations	4	2.38	1–4
Poor sleep	1	2.13	1–4

**Table 4 healthcare-11-00104-t004:** Correlations between emotional labor, emotional exhaustion, and physical and mental health.

Variable	Emotional Labor	Emotional Exhaustion	Physical and Mental Health
Emotional labor	1		
Emotional exhaustion	0.336 ***	1	
Physical and mental health	0.184 *	0.597 ***	1
Surface acting	0.918 ***	0.404 ***	0.135
Deep acting	0.659 ***	0.044	0.166
Physical condition	0.167	0.482 ***	0.857 ***
Anxiety	0.225 *	0.596 ***	0.878 ***
Depression and poor family relations	−0.070	0.294 ***	0.600 ***

*: *p* < 0.05; ***: *p* < 0.001.

**Table 5 healthcare-11-00104-t005:** The mediating effect of emotional exhaustion in the relation between emotional labor and physical and mental health.

Predictor	Physical and Mental Health		
	Model 1	Model 2	Model 3
Predictor			
Emotional labor	0.184 *		−0.019
Emotional exhaustion		0.336 ***	0.603 ***
$R^{2}$	0.184	0.336	0.597
△$R^{2}$	0.184	0.152	0.261
F	4.138 *	14.996 ***	32.447 ***

*: *p* < 0.05; ***: *p* < 0.001.

## Data Availability

Data is available upon request from the corresponding author. Data are not publicly available due to privacy and ethical constraints.

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
