# Peer review of "The Effect of Emotional Labor on the Physical and Mental Health of Health Professionals: Emotional Exhaustion Has a Mediating Effect"

_healthcare, 2022, doi:10.3390/healthcare11010104_

Round 1

Reviewer 1 Report

First of all, congratulations for the work done, then I will mention a number of changes and recommendations in order to obtain clearer and more accurate information.

- Comments on the introduction:

Lines 33 to 37 need references.

- Comments on results:

The results that talk about 1 ‰ and 5% are confusing, shouldn't they refer to 1% and 5%?

-Comments on conclusions:

I think you should put some limitations on your work, such as the limited sample size. It is not clear to me how many hospitals the study is carried out in, if only one, that would be another limitation.

Author Response

  1. Many thanks to the reviewer for the suggestion, references have been added.
  2. The statistical significance has been presented consistent.
  3. Supplementary research limitations. (Study limitations moved to end of Discussion paragraph as suggested by reviewer 2).

Reviewer 2 Report

Review Manuscript Healthcare-2013809

I find the manuscript fits the scope of Healthcare Journal. It includes research on emotional labor, emotional exhaustion and physical and mental health performed in health professionals, carried out in the period from August 2020 to July 2021.

Aspects affecting the whole manuscript:

I suggest authors use the term “health professional” instead of “medical professional” or “medical professional technicians” throughout the manuscript. In many countries medical refers to physician or medical doctors, but not to “clinical psychologists or nutritionists”. Because of this, I think that the word “health” is a better option than medical.

Abstract:

 Too long, quantitative results such as correlations should be in the results section. In the abstract authors should present the more important results only.

Change the first two sentences because they are repeated verbatim in the first four lines of the introduction. The change could be in the abstract or the introduction, change one or the other.

Do not specify the different health professionals in the abstract. Readers will find them in the method section later.

The methods section in the abstract seems to suggest that just a single questionnaire was used. Please include in this section just the variables measured and cite the analyses performed.

Introduction

I find it adequate but short

Method

Subsection Study Design:

This section should be named Study Design and Participants, since it is here where participants are described. Also, a good description of participants is needed in this subsection and not in the results section: list all type of health professionals included, how many females or males, age range, education, etc. Descriptive statistics about the sample should be here and not in the results section

Subsection Measures:

Authors should present the different questionnaires orderly and give a name to the questionnaire created, such as “emotional labor and exhaustion questionnaire”, giving all information about it (including content and reliability) the first time the questionnaire is presented. My advice about this questionnaire is that authors should perform at least a Factor Analysis asking for 3-4 factor.  This analysis should confirm the structure of three or four factors proposed.

It should be clear what variable is measure by each scale, and which of them are not used in the study. For example, in the questionnaire there is the variable “basic information” that is not used later. On the other hand, emotional exhaustion is measure by the authors` questionnaire and the Maslach Burnout Inventory, which of the two has been used in this study?

Information about the response scale in the basic information, emotional exhaustion, and mental health in the authors` questionnaire should be placed and added here.

Information about the different measures should be given the method section, measures subsection, and not in the results section.

Examples of items are needed in all the questionnaires used, but mainly in the emotional labor parts of “deep acting” and “surface acting”.

Reliability information of all variables should also be included when describing each questionnaire.

Data collection Subsection:

Better description is needed. I guess that the questionnaires are inside the “opaque envelope bag” given to participants, but it is better if the authors write it. Also, the word “impenetrable box” is not adequate, how the respondent introduced the completed questionnaire if the box is impenetrable. Better use the word “safe box” or “ballot box”.

Results

 3.1. Descriptive statistics of respondents’ basic information subsection

All section should be in the participants or respondent subsection in the method section. Included table 1.

3.2. Descriptive analysis of each scale subsection

This subsection should include the mean score in each variable and stipulate if they are high/medium/low. The remainder information should be in the subsection measures in the Method section.

The t-test and one-way ANOVA results should also be included at the end of the method section, to show that the demographic variables do not present differences in the variables studied. The quantitative no significant t-test and one-way ANOVA should also be included.

3.4. Differences in professional categories of emotional labor, emotional exhaustion, and physical and mental health subsection

The one-way ANOVA performed between the different health professional should be a Kruskal Wallis for independent samples analysis (a non-parametric analysis). I think that it is not possible for subsamples of 3 (audiologist?) or 8 (radiologist respiratory?) to present a normal distribution in all the variables. Therefore, a non-parametric ANOVA such as the Kruskal Wallis analysis is advisable.

Table 3. Please, revise the data, there are seven professional categories and the total, but nine lines with data. Also, place the numbers (n, means, SDs) on the right part of the table at the adequate level in the corresponding category. As it is presented now it is difficult to know which number correspond to each category.

Table 4. Change “poor family” by “poor family relations”.

Discussion

A paragraph about the limitations of this study is needed in this section

Conclusion

Do not present quantitative results in this section, just present the important conclusions

References

Some journal names are abbreviated and some are not. Authors should use the same criteria for all references

Reference numbesr 6, 7, 8, 10, 26, 28, the title of the book should be in italics not the editorial.

Correct the format of reference number 25.

Reference 31, “vocational” as a part of the journal name should be in capital letter.

Author Response

  1. Many thanks to the reviewer for the recommendation.
  2. Modified based on reviewer suggestions.
  3. The abstract has been revised as suggested by the reviewer.
  4. Thanks for reviewer comments
  5. Modified as suggested by reviewers to include participant background information in this paragraph.
  6. It has been modified according to the reviewer's suggestion, and the measurement tools are clearly described and included in the results of reliability analysis and validity analysis.
  7. Modified based on reviewer suggestions.
  8. Modified based on reviewer suggestions.
  9. Modified based on reviewer suggestions.
    • Divide the average scores of each variable into low, medium and high groups.
    • The t-test and One-way ANOVA analysis have been put at the end of the research method.
  10. Modified based on reviewer suggestions.
  • Use Kruskal Wallis analysis to analyze emotional labor, emotional exhaustion, and physical and mental health subsection of different professional categories.
  • Table 3 has been deleted.
  • Table 4 has corrected text
  1. Increased study limits as suggested by reviewers.
  2. Data has been removed as suggested by reviewers.
  3. Modified based on reviewer suggestions.

Round 2

Reviewer 2 Report

Manuscript HEALTHCARE-2013809. Review 2

ABSTRACT

Correct the following sentence: “used study with a purposive sampling.” It is wrong. Correct English.

METHOD

Measures

When authors write: “validity was 0.88” or “validity was 0.93.” or “validity was 0.89.” Which kind of validity are they referring? Content, concurrent, differential, social, ecological, etc…..

It has to be specified what kind of validity and how the specific validity has been obtained.

2.4. Differences in physical and mental health according to demographic characteristics

About the sentences: “Differences in physical and mental health due to gender and marital status were evaluated by a t-test, and there were no significant differences. One-way analysis of variance was used to explore the differences in physical and mental health according to age, education, and profession. There were no statistically significant differences found among the groups”

 It is necessary to write the results (quantitative), not only to say that there are not significant differences.

RESULTS

Statistical coefficients should be in italics (ex. P change by p, etc…)

When authors write: Cutoff scores for the above were as follows: low group < 1.35, middle group 1.35–3.65, high group > 3.65.

 It is necessary to explain the criterion that has been used for theses cutoff scores.

3.2. Differences in emotional labor, emotional exhaustion, and physical and mental health among health professionals

“Kruskal Wallis analysis was used to explore the differences in emotional labor (including, surface acting and deep acting), emotional exhaustion, and physical and mental health in the different professions. The results showed that surface acting (P < .05) and emotional exhaustion (P < .05) were significantly different among the health professional categories (F = 3.491, P < .05) (Table 3)”

 Authors should say which professionals obtained the higher or lower scores. Just saying that there are differences is not enough.

Table 3. The result for mental exhaustion should include the symbol “*”. This table could be deleted and include the results in the text. It looks like an empty table.

Table 4. All coefficients should include 3 digits.

3.3. The relationship between emotional labor, emotional exhaustion, and physical and mental health

All correlations are positive and most of them significant, the interesting part is why some of them are not significant. Authors could try to explain it.

 I do not see the relevance of correlations between emotional labor, and its two factors: surface acting and deep acting.

Remainder action asked for in the first revision not performed in the second one without explanation.

1. Authors could perform at least a Factor Analysis asking for 3-4 factors.  This analysis should confirm the structure of three or four factors proposed.

2. Examples of items are needed in all the questionnaires used, but mainly in the emotional labor parts of “deep acting” and “surface acting”.

Author Response

Thanks to the reviewer for the suggestion.

We have revised as suggested by the reviewer.

Reviewer’s Comments

Response to the Comments

ABSTRACT

Correct the following sentence: “used study with a purposive sampling.” It is wrong. Correct English.

Thanks for the reviewer's suggestion, it has been revised.

METHOD

Measures

When authors write: “validity was 0.88” or “validity was 0.93.” or “validity was 0.89.” Which kind of validity are they referring? Content, concurrent, differential, social, ecological, etc…..

It has to be specified what kind of validity and how the specific validity has been obtained.

Modified according to reviewer suggestions.

2.4. Differences in physical and mental health according to demographic characteristics

About the sentences: “Differences in physical and mental health due to gender and marital status were evaluated by a t-test, and there were no significant differences. One-way analysis of variance was used to explore the differences in physical and mental health according to age, education, and profession. There were no statistically significant differences found among the groups”

 It is necessary t write the results (quantitative), not only to say that there are not significant differences.

It has been modified according to the reviewer's opinion, and the statistical analysis data has been supplemented.

RESULTS

Statistical coefficients should be in italics (ex. P change by p, etc…)

When authors write: Cutoff scores for the above were as follows: low group < 1.35, middle group 1.35–3.65, high group > 3.65.

 It is necessary to explain the criterion that has been used for theses cutoff scores.

Thanks to the reviewer for the suggestion.

1. Corrected "P" to "p".

2. The score performance of each variable has been explained.

3.2. Differences in emotional labor, emotional exhaustion, and physical and mental health among health professionals

“Kruskal Wallis analysis was used to explore the differences in emotional labor (including, surface acting and deep acting), emotional exhaustion, and physical and mental health in the different professions. The results showed that surface acting (P < .05) and emotional exhaustion (P < .05) were significantly different among the health professional categories (F = 3.491, P < .05) (Table 3)”

 Authors should say which professionals obtained the higher or lower scores. Just saying that there are differences is not enough.

Table 3. The result for mental exhaustion should include the symbol “*”. This table could be deleted and include the results in the text. It looks like an empty table.

Thanks to the reviewer for the suggestion.

1. It has been supplemented to explain the differences between surface acting and emotional exhaustion among different health professionals (list higher and lower subgroups).

2. The original table 3 has been deleted.

Table 4. All coefficients should include 3 digits.

Modified according to reviewer suggestions.

3.3. The relationship between emotional labor, emotional exhaustion, and physical and mental health

All correlations are positive and most of them significant, the interesting part is why some of them are not significant. Authors could try to explain it.

 I do not see the relevance of correlations between emotional labor, and its two factors: surface acting and deep acting.

Thanks to the reviewer for the suggestion.

Correlations between variables have been supplemented to explain.

Remainder action asked for in the first revision not performed in the second one without explanation.

1. Authors could perform at least a Factor Analysis asking for 3-4 factors.  This analysis should confirm the structure of three or four factors proposed.

2. Examples of items are needed in all the questionnaires used, but mainly in the emotional labor parts of “deep acting” and “surface acting”.

Thanks to the reviewer for the suggestion.

I have supplemented the reliability of each scale, and also conducted factor analysis on the surface acting and deep acting of emotional labor.